# Rumba Dance Combined with Breathing Training as an Exercise Intervention in the Management of Stress Urinary Incontinence in Postmenopausal Women: A Randomized Controlled Trial

**DOI:** 10.3390/ijerph20010522

**Published:** 2022-12-28

**Authors:** Yuting Tang, Xian Guo, Yi Wang, Zeyao Liu, Guoxia Cao, Yanbing Zhou, Mengmeng Chen, Jingying Liu, Jinhao Mu, Mengjie Yuan

**Affiliations:** 1School of Art, Beijing Sport University, Beijing 100084, China; 2Space Science and Technology Institute (Shenzhen), Shenzhen 518038, China; 3Sport Science School, Beijing Sport University, Beijing 100084, China; 4Beijing Sports Nutrition Engineering Research Center, Beijing 100084, China; 5Physical Exercise Department, Renmin University of China, Beijing 100872, China

**Keywords:** pelvic floor dysfunction, exercise, dance, pelvic floor muscles, synergistic muscles

## Abstract

Purpose: Stress urinary incontinence (SUI) refers to involuntary leakage from the urethra, synchronous with exertion/effort, sneezing or coughing, which has a negative effect on quality of life. Studies have shown that mild-to-moderate physical activities reduce the risk of SUI by multiple mechanisms. The objective of this study was to determine whether the Rumba dance combined with breathing training (RDBT) can reduce the severity of incontinence and improve the quality of life of patients with SUI. Methods: A randomized clinical trial was conducted with women who were sedentary, were postmenopausal, reported mild-to-moderate SUI on a 1-h pad test, were not already engaged in Rumba dance and did not receive estrogen replacement therapy. The patients were randomly assigned to the RDBT group (*n* = 13) or the control group (*n* = 11). The intervention included 90 min of RDBT three times per week for 16 weeks, and the vaginal resting pressure (VRP), pelvic floor muscle (PFM) strength and endurance, 1-h pad test, International Consultation on Incontinence Questionnaire—Urinary Incontinence Short Form (ICIQ—UI SF), and the Incontinence Quality of Life Questionnaire (I—QOL) were measured or completed at baseline and 16 weeks. None of the participants reported adverse events. Results: The mean (±SD) age of the participants was 55.75 ± 5.58 years. After 16 weeks, in the RDBT group, the urine leakage on the 1-h pad test was significantly decreased −2.91 ± 0.49 from the baseline (*p* = 0.000). The VRP increased from 76.00 ± 16.23 cmH_2_O to 95.09 ± 18.90 cmH_2_O (*p* = 0.000), the PFM endurance of class I (−3.15 ± 1.99% vs. −0.46 ± 0.97%, *p* = 0.000) and class II (−0.69 ± 0.95% vs. −0.23 ± 0.44%, *p* = 0.065) increased, and the grades of PFM strength of class I and class II were significantly enhanced (*p* < 0.01). Finally, the severity of self-reported incontinence (ICIQ—UI SF) significantly decreased from 6.12 ± 2.15 to 3.81 ± 1.68 (*p* = 0.000), and quality of life (I—QOL) improved from 75.73 ± 11.93 to 83.48 ± 7.88 (*p* = 0.005). Conclusion: A 16-week RDBT program can increase PFM strength and endurance to reduce the severity of incontinence symptoms and improve the quality of life in patients with SUI, demonstrating the feasibility of recruiting and retaining postmenopausal women with SUI into a RDBT therapeutic program.

## 1. Introduction

Stress urinary incontinence (SUI), which is common among women [1,2], refers to a complaint of involuntary leakage due to effort or exertion or as a result of sneezing or coughing [3], and affects approximately 18.9% of Chinese women, with the age group of 50–59 years being the most affected at 28.0% [2,4,5]. Other research demonstrated that the odds and risks of incontinence increased with age [2] because of the progressive loss of estrogen after menopause, which is the most important cause of urogenital atrophy [6]. SUI increases the risk of stress, anxiety and depression, and thus has negative effects on women’s general health, well-being and quality of life [7,8].

According to the recommendation of the European Association of Urology, the treatment of urinary incontinence (UI) should start from conservative treatment and, above all, from physical therapy [9], such as pelvic floor muscle (PFM) training, weight management and promotion of physical exercise [9]. Many authors recommend PFM training to increase PFM strength and endurance during the rehabilitation of pelvic floor (PF) function [10,11]. However, research has indicated that the PF does not function as an independent entity [12], and recent literature supports the use of a global approach in motor control exercise programs [13]. In addition to the PFM, PF function is also supported by the pelvis [14] and other synergistic muscles, such as the diaphragm and transversus abdominis muscles [15,16,17]. Therefore, the pelvis and the synergistic muscles of the PFM should be taken into account in the therapeutic management of weakened PF. Many studies and reports have demonstrated the effects of exercise intervention on PF function. Mild-to-moderate physical activity decreases the odds and risks of urinary incontinence [10,18], that included PFM training, breathing training, vibration exercise, yoga, Pilates, and aerobic exercise (such as walking) [11,19,20]. Those physical activity were based on breathing control, increasing attention to the PFM, correcting body posture, and improving core physical exercise to activate the PFM and their synergists, and may induce positive effects on SUI through conscious and unconscious co-contraction of the PFM [18,21].

Although the above methods have shown potential, the threshold levels of frequency, intensity and duration are still unknown [21,22]. There are conflicting results [21,23]. Bo K suggested that there is not yet strong evidence that alternative exercise regimens can reduce urinary leakage in women with SUI [23]. Since the pelvis provides skeletal support for the pelvic floor, a change in pelvic orientation will affect the bioelectrical activity of the PFM [14]. Additionally, the bony and muscular pelvis is strongly connected to the gluteal muscles in terms of morphology and function; together, they provide support to the internal organs [15]. PFM contraction can be reinforced by the co-contraction of core and other muscles, such as abdominal muscles, hip muscles, and adductor muscles, causing increased PFM tension [16,17]. Thus, maintenance of the normal physiological function of pelvic organs depends on the close cooperation and interaction among the pelvis, PFM and their synergists [12]. Hence, alternative, feasible and appropriate complementary interventions also deserve attention when considering the treatment and prevention of SUI.

Rumba dance focuses on pelvic and spinal stability and mobility, it requires a certain amount of control, strength and flexibility to complete the movement [24]. Almost all of the basic movements in Rumba dance are performed with contraction of abdominal muscles, hip muscles, and respiratory muscles [25]. The improvement of those muscles may promote the function of pelvic floor [15,16,17]. Meanwhile, the Rumba dance focuses particularly on the coordination of breathing and movement. Proper breathing exercise not only activates the PFM and their synergists but also adjusts the intra-abdominal pressure [26,27]. The PF, abdomen and diaphragm act in a coordinated manner to form the trunk stabilization system, which provides stability to the abdominal-lumbo-pelvic cavity by adjusting the intra-abdominal pressure [27].

The effects of RDBT as an exercise intervention for SUI have not been examined, we aims to determine whether RDBT can reduce the severity of incontinence and improve the quality of life for postmenopausal patients with SUI.

## 2. Methods

### 2.1. Study Design

This randomized controlled trial was conducted at the laboratory of Women’s Sports and Health Research, Beijing Sport University, China (Figure 1). The study was approved by the Sports Science Experimental Ethics Committee of Beijing Sport University (2021054H). The laboratory of Women’s Sports and Health Research, Beijing Sport University, carried out web-based randomization, with all eligible participants assigned in a ratio of 1:1 to either the RDBT group (*n* = 15) or the control group (*n* = 15). Randomization was based on the severity of urinary incontinence, the number of births (one or two times), and the number of vaginal deliveries (one or two times) to prevent any effect on outcomes of an uneven distribution over the group allocation. Group allocation was relayed to participants by WeChat. Participants and study personnel involved in administering interventions could not be masked to group allocation. However, clinicians who performed the PFM assessment at baseline and 16 weeks were masked.

### 2.2. Sample and Participants

A total of 86 sedentary postmenopausal women (45–65 years old) volunteered to participate in this randomized controlled trial, and 56 volunteers who did not meet the screening criteria were excluded. A total of 30 women were randomly allocated to an intervention or a control group. Investigators contacted the people who reported symptoms of urinary incontinence after completing the basic situation survey by telephone and conducted a preliminary screening by asking them whether they suffered any emergency leakage or abdominal pressure leakage, leakage frequency, or any symptoms associated with leakage. The type and severity of participants’ urinary incontinence were confirmed by the investigators in reference to the urinary leakage symptoms, 1-h pad test and International Consultation on Incontinence Questionnaire–Urinary Incontinence Short Form (ICIQ–UI SF) outcomes. Recruitment was carried out through WeChat and delivery of leaflets in the Haidian District residential area of Beijing.

Participants were included if they met the following inclusion criteria: (1) mild-to-moderate SUI (self-reported involuntary urine leakage on effort, exertion, sneezing or coughing, ICIQ—UI SF score < 13, and leakage on the 1-h pad test of 1–2 g (mild) or 2–10 g (moderate)), (2) sedentary behavior (physical activity ≤ 3 days/week and exercise duration ≤ 20 min/session [28]), (3) postmenopausal (at least 12 months since the last period), (4) vaginal delivery of singleton/multiple births, (5) no contraindications for physical exercise (evaluated by the Physical Activity Readiness Questionnaire (PAR-Q)), and (6) able to perform the prescribed exercise or exercise testing associated with the study. We excluded (1) women who had severe SUI (ICIQ—UI SF score ≥ 13 and leakage on 1-h pad test ≥ 10 g), (2) women who received hormonal replacement therapy (i.e., estrogen) or participated in other rehabilitation programs for urinary incontinence, (3) woman who has taken part in Rumba dance exercises (4) women who were unable to contract their PFM on request during a pelvic floor function examination, and (5) women who were unable to perform the tests and training sessions. All participants completed the PAR-Q and provided written informed consent.

### 2.3. Procedure

All participants were taught the correct technique for contraction of the PFM in preparation for the PFM assessment. The Rumba dance prescription was based on level-1 in the Technique Book of Chinese Dance Sport Federation, including the basic movements Change Weight in Place, Time Step, Cucaracha, Basic Movement, and Forward and Backward Walk. We used surface electromyography to test the muscles involved in the different movements, so that we could determine which were the most effective movements for activating certain muscles. Surface electromyography showed that all of those basic movements can significantly activate the gluteus maximus, rectus abdominis and adductor muscles. The participants in RDBT group engaged in 90 min of RDBT three times per week for 16 weeks. The control group did not receive any exercise or nutritional intervention. Every two weeks, a call or WeChat return visit was carried out with the participants and emphasized maintaining their daily physical activity and eating habits.

### 2.4. Intervention

Each session was divided into two parts. The first part was a 20-min abdominal breathing exercise, and the second part was a 60-min session of Rumba dance training. Additionally, there were five minutes of warm-up and five minutes of relaxation. To control the adherence to exercise, attendance was recorded at the beginning of each session, and home-based exercise was requested for those who did not attend. We ensured that all participants completed at least 90% of the course. All participants received documents through the WeChat group, including the methods of the abdominal breathing exercise, the pathogenesis of stress urinary incontinence and the anatomy of the pelvic floor.

Breathing exercises were performed in the lying position, and the participants were given the following instructions: Put one hand on the abdomen and the other on the chest. Inhale through the nose, then bulge the abdomen outward without chest undulation. Hold the breath for five seconds, and then exhale slowly through the mouth for five seconds. That process lasted for 10 min and was supplemented with blow-ups, using balloons of 5–30 cm, to increase the intensity of the exercise. When inspiring for 5 s, the two sides of the ribcage open outward, the abdomen rises upward, and the PFM relax. When blowing-up the balloon for 5 s, the two sides of the ribcage retract inward, pushing the abdomen back forcefully, and the PFM contract and resist the abdominal retraction. This process also lasted for 10 min. Participants were instructed to pay attention to the activity of the PFM during the breathing exercises to better identify and control the PFM. At the same time, participants were expected to pay attention to the activity of the pelvis and abdominal muscles to be better able to use the pelvis, PFM, and abdominal muscles during Rumba dance.

In addition to the breathing exercise, participants also underwent sessions of group-based Rumba dance exercise three times per week for 16 weeks. Rumba dance included 6 basic movements: change weight in place, time step, basic movement, basic forward movement, basic backward movement, and cucaracha. Each session was divided into three parts: (1) The first part was warm-up activities, including basic head, shoulder, chest, stride, knee, and ankle movements, and this process lasted for 5 min. (2) The second part was the exercise of basic movement. This part adopted the teaching method of a, a + b, a + b + c, which added a new basic movement based on the previous section. This process lasted for 60 min. (3) The third part was relaxing exercise through proper muscle stretching, and this process lasted for 5 min.

### 2.5. Outcomes

Outcomes were measured at baseline and after 16 weeks of intervention. The primary outcome measure was the 1-h pad test result, and the secondary outcome measures were pelvic floor function, the ICIQ—UI SF score and the I—QOL score. The evaluation of PFM strength and endurance was performed by the same clinicians.

A 1-h pad test, which is a standardized quantitative method for urine loss, was used to assess the symptoms of urinary incontinence through the amount of urine lost [29]. Participants put on a preweighed pad, drank 500 mL water in 15 min, and then performed a series of activities, including walking for 30 min, going up and down stairs, coughing, running, and others. Afterward, the pad was weighed again to measure the amount of urine leakage; 1–2 g leakage is considered mild, and 2–10 g leakage is considered moderate.

Pelvic floor muscle function was measured using a neuromuscular system (PHENIX USB 4, Vivaltis, Montpellier, France), the same equipment as used in Navarro Brazález B’s study [30]. Vaginal resting pressure (VRP) during contraction of the PFM was measured using a manometer connected to a sensor, which was inserted into the vagina. The normal values were as follows: 80–150 cmH_2_O for VRP, 0–5 for PFM strength (PFMS was classified as grade 0 if it reached 40% of maximum strength within 10 s and was highly persistent for 0 s, it was classified as grades 1, 2, 3, 4, and 5 if it was persistent for 1, 2, 3, 4, and ≥5 s, respectively.) and 0% for endurance (the endurance indicates how long the pelvic floor muscles continue to contract, negative or positive values indicate decreased PFM endurance). PFM is divided into Class I and II muscle fibers. Class I refers to the slow muscle fibers, which belong to the deep muscle group and mainly to maintain the rich and strong organs in the normal position. Class II refers to the fast muscle fibers, which belong to the superficial muscle group and the main function is to control urine and sexual quality of life.

For the severity of urinary incontinence (assessed with ICIQ—UI SF), the scores, which ranged from 0–21, were calculated from the weighted sum of three items, including urinary incontinence frequency (“how often do you leak urine?” 0 = never to 5 = all the time), leakage quantity (“how much urine do you usually leak?” 0 = none to 6 = a large amount), and interference with everyday life (0 = not at all to 10 = a great deal), with higher scores reflecting greater severity. That questionnaire also included a fourth ungraded item to assess patients’ perceptions of the cause and type of leakage. The impact of urinary incontinence on quality of life (assessed with I—QOL) was scored from 0–100, and higher scores reflected a lower effect of incontinence on quality of life.

### 2.6. Statistical Analysis

Statistical analyses were undertaken using IBM SPSS Software (v.25). Participant characteristics were summarized with counts (percentages) for categorical variables and means (standard deviations) for continuous variables. The independent *t* test and chi-square test were used to compare groups regarding the general characteristics at baseline of continuous variables and categorical variables. Within-group differences between continuous and categorical variables were evaluated using a repeated measure design and chi-square test. Statistical significance was established as *p* ≤ 0.05.

## 3. Results

After 16 weeks, two participants in the RDBT group and four participants in the control group withdrew because the corona virus disease 2019 (COVID-19) (Figure 1). The total sample consisted of 24 women aged between 45 and 65 years, with a mean age of 55.00 (SD = 5.85) years. Participant individual characteristics and urinary incontinence symptoms were similar between the groups at trial entry (Table 1). There were not any statistically significant differences in age, height, number of births, number of vaginal deliveries, or 1-h pad test severity between the missing participants and valid participants.

### 3.1. Leakage on the 1-h Pad Test

The primary outcome, leakage on the 1-h pad test at 16 weeks, was significantly decreased −2.91 ± 0.49 from the baseline (*p* = 0.000) in the RDBT and was not significantly different in the control group (*p* = 0.632) (Figure 2).

### 3.2. Function of PFM

The VRP in the RDBT group significantly increased from 76.00 ± 16.23 cmH_2_O to 95.09 ± 18.90 cmH_2_O (*p* = 0.000), but there were no significant differences in the control group (76.00 ± 17.34 vs. 68.73 ± 20.84, *p* = 0.073) (Figure 3). The PFM endurance of class Ⅰ (−3.15 ± 1.99% vs. −0.46 ± 0.97%, *p* = 0.000) and class Ⅱ (−0.69 ± 0.95% vs. −0.23 ± 0.44%, *p* = 0.065) in the RDBT group was increased, and there were no significant changes in class Ⅰ (−2.55 ± 2.07% vs. −2.18 ± 1.54%, *p* = 0.527) and class Ⅱ (−1.09 ± 0.70% vs. −1.18 ± 1.08%, *p* = 0.728) in the control group (Figure 4). The grades of PFM strength of class Ⅰ and class Ⅱ in the RDBT group were significantly enhanced (*p* < 0.01), and no statistically significant changes were observed in class Ⅰ and class Ⅱ in the control group (Table 2).

### 3.3. Severity of Urinary Incontinence

The severity of self-reported ICIQ—UI SF in the RDBT group was significantly reduced from 6.12 ± 2.15 to 3.81 ± 1.68 (*p* = 0.000), but there were no significant differences in the control group (from 5.36 ± 1.60 to 6.05 ± 1.74, *p* = 0.215). All participants completed the questionnaire again 7 days after first completing the questionnaire, and the test-retest reliability showed adequacy (0.792 at baseline and 0.937 at 16 weeks) (Figure 5).

### 3.4. The Quality of Life Score

The quality of life (I—QOL) in the RDBT group improved from scores of 75.73 ± 11.93 to 83.48 ± 7.88 (*p* = 0.005) and from 76.32 ± 8.87 to 73.50 ± 10.68 (*p* = 0.314) in the control group. All participants also completed the questionnaire again 7 days after the first completion, and the test-retest reliability showed adequacy (0.717 at baseline and 0.827 at 16 weeks) (Figure 6).

## 4. Discussion

This study provided preliminary evidence of a reduction in the severity of SUI after the 16-week RDBT program, and our findings showed a range of benefits for women with mild-to-moderate SUI who participated in the RDBT program, including improved PFM strength and endurance, decreased severity of incontinence and increased quality of life. Based on the movement characteristics of Rumba dance, the mechanism of SUI and the anatomy of the pelvic floor, there are several reasons why the RDBT program had a positive effect on SUI.

Previous researches found the condition of SUI is associated with age, estrogen level and high BMI, especially the accumulation of abdominal fat, which increases pressure on the pelvic floor [5,31,32]. However, other studies proved that persistent Rumba dance may help to prevent SUI from developing by reducing body weight and slowing the reduction in estrogen levels with aging [33,34,35,36,37].

In addition, the most distinctive feature of Rumba dance is the charming swing and twist of the hip during the Rumba dance, and the most common movement pattern of the hip is a figure of “∞” that is twisted and traversed in cross-section [25,38]. The hips can move along three planes in Rumba dance, including traverse and tilt on the frontal plane, twist on the transverse plane, and forward and backward movement on the sagittal plane [25,38]. Each hip movement is a combination of one or more movements of the entire pelvic structure [25]. The pelvis serves as a skeletal support for the entire pelvic floor and the pelvic cavity [39]. PFM are the muscular layer of the pelvic floor; their function is to support the pelvic organs and contribute to the closure of the urethra and anus [40].

It is unclear whether there is a direct or indirect connection between the pelvis orientation and the PFM. Ptaszkowski K et al. [14] have shown a direct relationship between the orientation of the pelvis and PFM activation, and that movement of the pelvis on the sagittal plane increases the resting and functional bioelectrical activity of the PFM. As a result of pelvis orientation affecting PFM activation, Rumba dance is likely to activate the PFM by hip swing and twist. Physical therapy for SUI mainly focuses on improving the muscle strength of the PFM [9,11]. The findings of the present study suggest that the VRP and the strength of the PFM were significantly increased after 16 weeks of RDBT program intervention. Notably, the PFM strength of class I increased, indicating that in addition to improving the shallow muscle strength of the PF, RDBT also had a positive effect on the deep PFM.

In addition to the involvement of the PFM in the movement of the pelvis, RDBT also involves a large number of the other core muscles, such as the abdominal muscles, the muscles of the hip, the gluteus maximus muscles, and the adductor muscles [25,41], which are synergists of the PFM through myofascial continuity with the pelvic floor [12,39]. Those muscles are functionally and morphologically connected to the pelvis and indirectly to the PFM [42] and play a vital role in the stability of the center of gravity and the support of the pelvis. Those muscles are used extensively, especially in the weight transfer and movement of the pelvis in dance [43] and to control posture and stabilize the spine during weight transfer in Rumba dance [25,41]. Other research has proven that the activation of these synergistic muscles is an important conservative treatment for SUI [15,16,17,42,44,45,46]. Rumba dance is an entertaining sport with music, and almost every dance movement is accompanied by a large amount of hip and pelvic rotation traverse, twist and tilt. As the hips and pelvis undergo repetitive motion, the center of gravity changes, and the PFM and other core muscles actively participate in the dance [41,43].

However, posture control, spinal stabilization and breathing are mechanically and neuromuscularly interdependent, and they all involve abdominal wall muscle and PFM [47]. Breathing exercises also stimulate PFM activation, especially during forced exhalation, PFM contraction and maintaining the position of the pelvis [48]. Several studies suggested that abdominal breathing can lead to reflex activation of muscles in the abdominal wall and pelvic floor, thereby reducing the severity of urinary incontinence [49,50,51]. In this study, the abdominal breathing exercise was used to train respiratory function and supplemented with balloon-blowing training to increase the intensity and stimulation of respiration; moreover, body movement needs to be combined with breathing in Rumba dance, and the breathing is controlled, thus increasing respiratory function. The pelvic floor muscles formed at the floor of the abdominal cavity contribute to the control of intra-abdominal pressure.

This was the first study to explore the effects of the RDBT program on stress incontinence in postmenopausal women. Future research is needed to further explore the possible mechanism of the RDBT program effects on urinary incontinence, which could make the treatment and prevention of urinary incontinence a pleasant exercise experience.

### Limitations and Future Research

To the best of our knowledge, this is the first randomized controlled trial to evaluate the effectiveness of Rumba dance combined with breathing training in women with SUI, and it has shown positive improvement effects. However, this study has several limitations.

One possible limitation is the small sample size due to the COVID-19 pandemic, with various unpredictable limitations in recruitment, data collection, and exercise interventions, which added a challenge to the study. Due to the small sample size, the independent effect of the Rumba dance on SUI could not be determined. Further studies with larger sample sizes will help bolster the results of this study. Another limitation was that the assessment methods for UI type and severity in regard to the urinary leakage symptoms were the 1-h pad test and ICIQ–UI SF rather than urodynamic and clinical assessment.

The pelvic floor consists of muscle, ligaments and fascia and is connected to distal areas of the body (such as the foot) through myofascial connections [52]. Due to tissue continuity, functional disorder of ligaments and fascia (even areas away from the pelvic floor) may lead to pelvic floor disorder [53]. They are also important for promoting body consciousness and maintaining body stability in Rumba dance [54]. In the dance process, the dancer not only fully stretches the movement to reflect the flexibility of the body but also maintains a certain degree of tension, reflecting the balance between hardness and softness. With the constant practice of Rumba dance, the elasticity of the fascia and ligaments increase. This study investigated the effect of Rumba dance on PFM strength, but the flexibility of the PF was not discussed. Further research is needed to explore the long-term efficacy and the mechanism of action of this intervention. Rumba dance may be an effective treatment option for SUI.

## 5. Conclusions

A 16-week RDBT program can increase PFM strength and endurance to reduce the severity of incontinence symptoms, then it can improve the quality of life in patients with SUI, demonstrating the feasibility of recruiting and retaining postmenopausal women with SUI into a RDBT therapeutic program.

## Figures and Tables

**Figure 1 ijerph-20-00522-f001:**
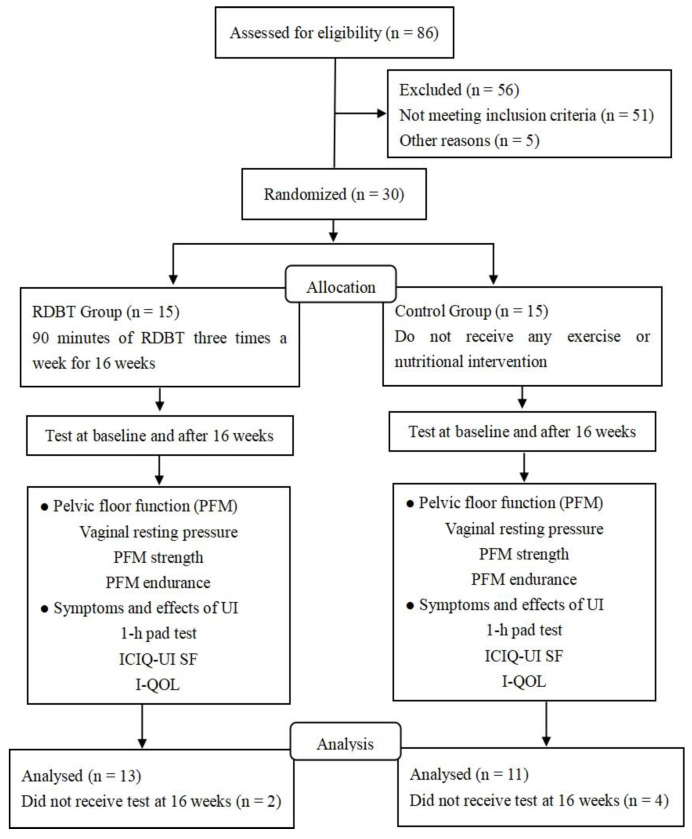
Flowchart of the study.

**Figure 2 ijerph-20-00522-f002:**
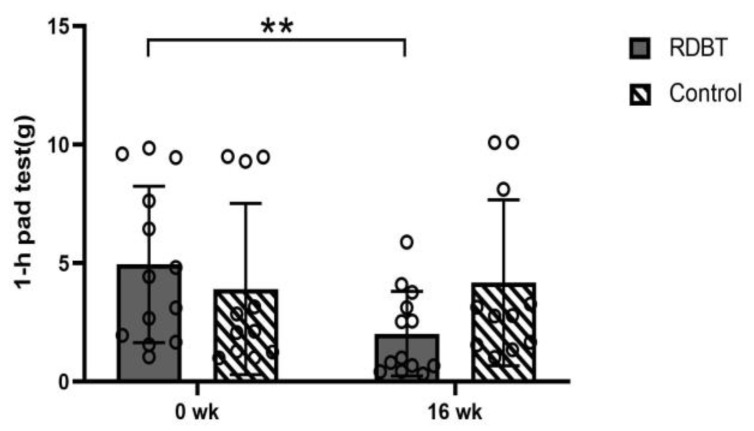
Leakage on the 1-h pad test. The circles represent the individual values. ** *p* < 0.01 was considered statistically significant between baseline and postintervention in the RDBT group.

**Figure 3 ijerph-20-00522-f003:**
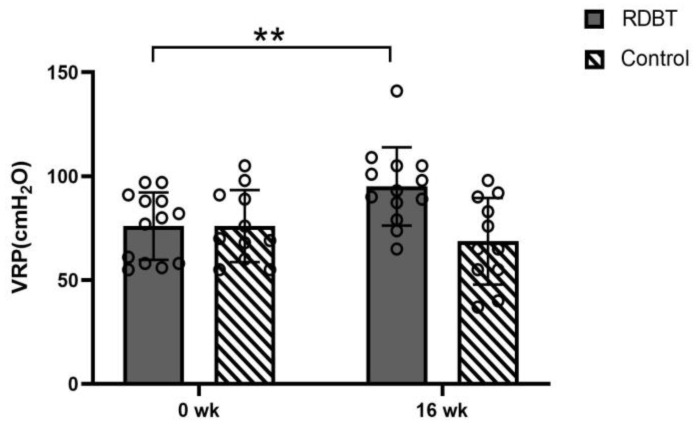
Changes in vaginal resting pressure (VRP). The circles represent the individual values. ** *p* < 0.01 was considered statistically significant between baseline and postintervention in the RDBT group.

**Figure 4 ijerph-20-00522-f004:**
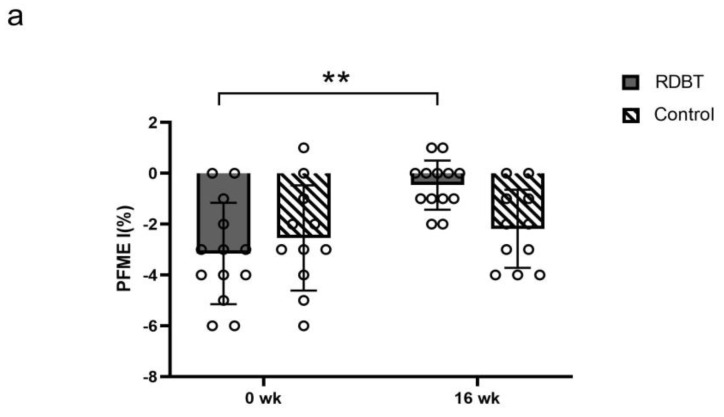
Percentage changes in pelvic floor muscle endurance (PFME, indicating how long the pelvic floor muscles continue to contract) of class I (**a**) and class II (**b**). The circles represent the individual values. ** *p* < 0.01 was considered statistically significant between baseline and postintervention in the RDBT group.

**Figure 5 ijerph-20-00522-f005:**
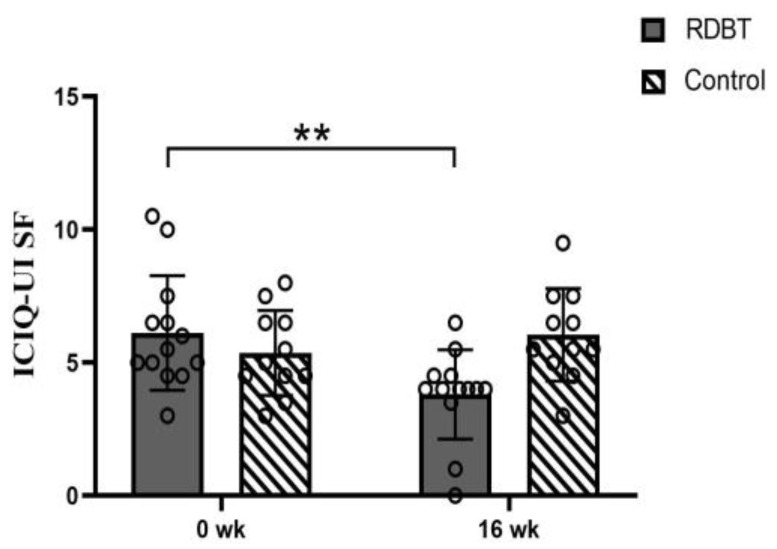
Change in the ICIQ–UI SF score. The circles represent the individual values. ** *p* < 0.01 was considered statistically significant between baseline and postintervention in the RDBT group.

**Figure 6 ijerph-20-00522-f006:**
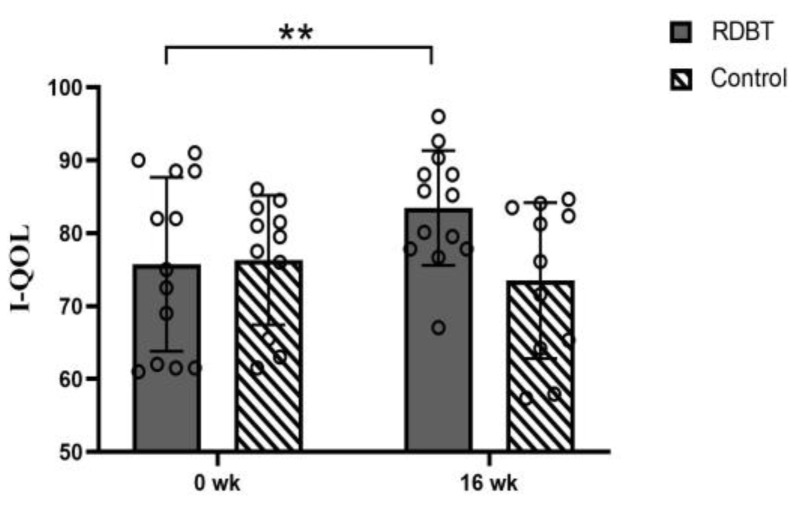
Change in the I—QOL score. The circles represent the individual values. ** *p* < 0.01 was considered statistically significant between baseline and postintervention in the RDBT group.

**Table 1 ijerph-20-00522-t001:** Baseline characteristics of participants assigned to the RDBT group and control group. Values are numbers (percentages) unless stated otherwise.

Characteristics	RDBT Group	Control Group	*p* Values
Age (years) mean (SD)	53.23 (5.79)	57.09 (5.45)	0.919
Height (cm) mean (SD)	161.49 (4.28)	161.12 (6.57)	0.261
BMI (kg/m^2^) mean (SD)	23.83 (2.92)	23.27 (2.87)	0.638
Number of births			0.902
1	92.3% (12/13)	91% (10/11)	
2	7.7% (1/13)	9.0% (1/11)	
Number of vaginal deliveries			1.000
1	92.3% (12/13)	91% (10/11)	
2	7.7% (1/13)	9.0% (1/11)	
1-h pad test severity			1.000
Mild (leakage < 2 g)	30.8% (4/13)	36.4% (4/11)	
Moderate (2 g ≤ leakage < 10 g)	69.2% (9/13)	63.6% (7/11)	

**Table 2 ijerph-20-00522-t002:** Changes in PFM strength baseline and post-intervention in both groups.

Group		Baseline		Post-Intervention	
		<3 (Grade)	≥3 (Grade)	<3 (Grade)	≥3 (Grade)
PFMS I					
	RDBT	38% (5/13)	62% (8/13)	0% (0/13)	100% (13/13) **
	control	18% (2/1)	82% (9/11)	18% (2/11)	82% (9/11)
PFMS II					
	RDBT	54% (7/13)	46% (6/13)	0% (0/13)	100% (13/13) **
	control	18% (2/11)	82% (9/11)	18% (2/11)	82% (9/11)

Changes in pelvic floor muscle strength (PFMS) of class I and class II. “<3” represents the grades from the 0–2 scale, “≥3” represent the grades from the 3–5 scale. ** *p* < 0.01 were considered statistically significant for within-group postintervention compared with baseline.

## Data Availability

The data presented in this study are available on request from the corresponding author. The data are not publicly available due to privacy or ethical restrictions.

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
