# Peer review of "Rumba Dance Combined with Breathing Training as an Exercise Intervention in the Management of Stress Urinary Incontinence in Postmenopausal Women: A Randomized Controlled Trial"

_ijerph, 2022, doi:10.3390/ijerph20010522_

Round 1

Reviewer 1 Report

The authors have taken a very important aim to determine whether Rumba dance can reduce the  severity of incontinence and improve the quality of life of patients with stress urinary incontinence. This is especially important in the population of postmenopausal women, in whom pelvic floor dysfunction is common.

This manuscript is generally well written, in intelligible language, and follows the content structure required for scientific manuscript.

However, I do suggest some changes and corrections, as follows:

Lines 107 “Thirty middle-aged and elderly women (45–65 years old) volunteered to participate. I think you should write here that 86 volunteered to participate, which you could asses using the eligibility criteria (what you presented in the Figure 1).

Line 122: change “your last period” for “the last period”

Lines 178-183: I suggest to delete the following part:  “At the 4th, 8th 178 and 12th weeks after the intervention, 3 participants were randomly selected to perform  the surface electromyography test of core muscles in the Rumba dance basic movement; their results were used as a reference to ensure that each participant correctly implemented the exercise plan, to monitor the training results, to determine the problems in the training process and to give effective feedback.” Later you don’t use the data from these assessments. If you consider it is important, please present the data in the Result section and discuss it in the Discussion section (e.g. in which Rumba movements which muscles were mainly involved).

Figure 1: change the box “Test at baseline and 16 weeks” for “Test at baseline and after 16 weeks”

Line 226: indicate that 5.85 expresses the standard deviation

Table 1: please present the p values for the comparison between groups (even the outcomes are not statistically significant). For your consideration: it should be valuable to present exact p-value for each outcomes, not just like the p < 0.01. The exact p-value can be used later by other authors, e.g. to estimate the sample size for their studies.  

Line 244 - 245: Please explain in the Methods section and in the Figure 4 what you meant using the terms “the endurance of class I” and  “the endurance of class II”.

Table 3: Please explain what you meant using PFM I and PFM II; please also explain what the “3” expresses (I guess scores from the 0-5 scale). All explanations should be in footnotes, which makes it easier for the reader to understand without having to go back to the Methods section.

Please use space between before and after “±” , e.g. 95.09 ± 18.90 throughout the text.

Despite the indicated points to be corrected or supplemented, I believe that this study is of great practical importance. The results highlight the need to promote various exercise programs for postmenopausal women, including Rumba classes, to reduce the incidence and severity of stress urinary incontinence in this population. After a proper revision of this manuscript, it should definitely be published.

Author Response

Dear reviewer,

  We deeply appreciate your valuable comments concerning our manuscript. The suggestions are all very helpful for the revision and improvement. We have carefully studied these comments and made related modifications, which we wish could meet with your approval.

  The point-by-point responses to the comments and main corrections in the manuscript are listed in the attachment.

  We would like to express our gratitude for your guidance on how to improve the quality of our study. We hope that the revised manuscript is now suitable for publication.

  Thanks again for all your time and effort.

  Kind regards.

Reviewer 2 Report

This is an interesting and well conducted study of a pelvic floor exercise intervention. However, the title and discussion is misleading as the intervention includes pelvic floor education and exercise as well as Rumba and the control group has not been chosen to assess the effect of Rumba. The results might very well be due to the pelvic floor or general exercise rather than the dance. Please find below some brief points requiring clarification and some discussion points to consider:

Line 101: I’m not sure what is meant by ‘randomization was minimized by’

Line 107: I don’t think you can call women 65 and under ‘elderly’!

Line 125: Why did you exclude severe SUI?

Figure 1 is misleading – the arrows do not make sense and the split list of test suggests the control and rumba group have different testing

Table 1 should include BMI rather than height

Section 3.2: Class I and Class II group are not explained. And what is the reason this arbitrary designation of <3 and >+3?

The paper is somewhat misleading, since it suggests this is a test of Rumba dance, yet a significant part of the intervention is pelvic floor education, focus and breathing. Despite much speculation on the specificity of Rumba for pelvic floor exercise, due to your choice of control (i.e. no intervention) this study is not designed to test Rumba specifically and changes can just as easily be attributed to the pelvic floor education and exercises, or an increase in exercise in general. The discussion and conclusion should be modified to reflect this limitation of the study design.

Since no information regarding the experience of the participants in the intervention is given, the conclusion cannot demonstrate feasibility of the intervention.

Author Response

(The authors gave the same response as above.)
